# Nanocomposite Synthesis of Nanodiamond and Molybdenum Disulfide

**DOI:** 10.3390/nano9070927

**Published:** 2019-06-27

**Authors:** Youngjun Kim, Dukhee Lee, Soo Young Kim, Eunah Kang, Chang Keun Kim

**Affiliations:** School of Chemical Engineering and Material Science, Chung-Ang University, Seoul 06974, Korea

**Keywords:** nanocomposite, MoS_2_, nanodiamond

## Abstract

A chemically conjugated nanodiamond (ND)/MoS_2_ nanocomposite was synthesized with amine-functionalized MoS_2_ and acyl chloride-coordinated ND. The chemical structure and morphology of the nanocomposite were characterized to examine the dispersion of MoS_2_ on the ND platform. The results revealed that the degree of dispersion was enhanced with increasing ratio of MoS_2_ nanosheets to ND. Moreover, the nanosheets consisted of several molecular interlayers that were well-dispersed on the ND platform, thereby forming a nanophase. The efficient electrocapacity of the ND/MoS_2_ nanocomposite was considerably greater than that of the MoS_2_ electrode alone. Furthermore, the nanophase distribution of MoS_2_ on ND with a graphitic shell provided a large surface area and reduced the diffusion distance of ions and electrons. Therefore, the nanophase electrode showed higher electrochemical capacitance than that of the MoS_2_ electrode alone.

## 1. Introduction 

Numerous novel materials and composites including two-dimensional (2D) materials have been exploited with the aim of enhancing electrocapacity, which can be applied for biosensing platforms and electrocatalytic performance. A 2D layered material can be defined as an unsupported crystalline solid with molecular layer thickness characterized by intralayer storage for heat, charge, and light transport [1,2,3,4]. This transport occurs in the presence of intralayer covalent bonds and intercalation-based interaction [5]. Specifically, MoS_2_ as a 2D transition metal dichalcogenide (TMD) exhibits the unique characteristic of charge confinement in the 2D layer in the absence of interlayer interaction along the z-axis [6]. Functional features of 2D MoS_2_ layers include high thermal and chemical stability for functionalization [7,8], large surface area [4], good mobility [9], and intercalation-based physical interaction [10,11]. For final functional performance, the construction of an electrode and nanophase distribution of an aggregation-free 2D MoS_2_ molecular layer are critical to induce maximal functional features. 

Carbon-based materials including fullerenes, graphene, and carbon nanotubes, are considered as fundamental platform materials of energy conversion and storage, owing to their thermal stability, conductivity, and mechanical properties [12,13,14,15]. Furthermore, the electron buffering capability of these materials stems from their high surface-to-volume ratio and the unsaturated carbon ᴨ bonds. Physicochemical characteristics can reduce oxygen adsorption on a catalyst surface, thereby improving the material performance. In general, the porous structure of carbon materials is characterized by a multiscale nanocage and a high surface-to-volume ratio that provide electron transfer and electrocatalytic active sites. These advantages have often been applied to nanocomposites with metal to induce synergistic functional effects, including solar cells, batteries, and supercapacitors [16,17,18,19,20]. 

Nanodiamonds (NDs) among carbon platforms have received less attention than some of the other carbon materials due to the high costs and low conductivity of these materials as electric functional materials [21,22]. However, NDs possess excellent physical properties, including high adsorption [23], high surface-to-volume ratio [24], chemical modality on the graphitic shell surface [25], and a nanocage within agglutinate [26]. Moreover, the dispersive capability of NDs in aqueous environments makes it possible to integrate and to extend materials to other functional employment, including biomedical applications as well as electronic functions [27]. NDs serve as a supporting platform that can augment the functional efficiency of the other incorporated pair of 2D materials. 

Therefore, in this study a ND/MoS_2_ nanocomposite by subsequent functionalization was synthesized and characterized. An ND with a graphitic shell around its surface provides stable sites for chemical modification. Immediate chemical reaction between the acyl chloride of NDs and amine-functionalized MoS_2_ was intended to form aggregation-free MoS_2_ nanosheet dispersion on the ND support (Scheme 1). This nanophase dispersion of MoS_2_ in several interlayers on the ND platform was examined from the viewpoint of physical characteristics and potential electrofunctionality. 

## 2. Experimental Methods

### 2.1. Nanocomposite Formation of MoS_2_ and Nanodiamonds

ND-COOHs (40 mg) gifted from Nanoresource (Seoul, Korea) were mixed with 100 mL of thionyl chloride (Samcheon Chemical Co., Seoul, Korea) and 0.5 mL of dimethylformamide anhydrous (DMF) in a 250 mL round-bottomed flask. The ND-COOHs were well-dispersed for 15 min of bath sonication under an ice bath. The ND dispersion for acylation was stirred for 24 h at 70 °C under a N_2_ purge. After reaction, the dispersion was washed repeatedly five times with tetrahydrofuran anhydrous (THF) and the NDs, separated by centrifugation, were dried at 60 °C in an air-circulated oven. 

For exfoliation, the mixture of pristine MoS_2_ (120 mg, <2 μm; Sigma-Aldrich Chemicals, St Louis, MO, USA) and N-vinyl-2-pyrrolidone (60 mL) was sonicated with condition of 55% power amplitude and 3 s pulse (VC750, Sonics Vibra-cell) for 8 h under circulation at 7 °C [28]. After exfoliation, the large MoS_2_ was separated by centrifugation (4000 rpm, 5 min), and the supernatant was extracted. This supernatant was then filtered through a syringe filter (0.45 µm, HP045AN, Advantec, Taipei, Taiwan) and centrifuged (14,600 rpm, 10 min). Afterward, the settled MoS_2_ nanosheets were washed with isopropanol and centrifuged repeatedly until the yellow color of N-vinyl-2-pyrrolidone became colorless. The nanosheets were then dried for 24 h at 40 °C under vacuum.

The MoS_2_ nanosheets (1 mg/mL) were dispersed in DMF. For chemical conjugation, 200 μL of cystamine (5 mg/mL) in DMF was added to 1 mL of the MoS_2_ dispersion. This dispersion mixture was then sonicated for 1 h under an ice bath. After sonication, the mixture was left to stand at room temperature for 24 h. Then, functionalized MoS_2_ was washed repeatedly three times with DMF, separated by centrifugation at 14,600 rpm, and completely dried at 60 °C in a vacuum oven. 

Acylated NDs (1 mg/mL) and amine-functionalized MoS_2_ nanosheets in DMF were dispersed with 5 min of bath sonication. The dispersion of acylated ND and amine-functionalized MoS_2_ with desired ratios (1:1, 1:2, 1:4, and 1:8) was mixed by sonication under an ice bath for 30 min, and shaken with a vortex for 24 h. After the reaction, the composite was washed and completely dried at 60 °C in a vacuum oven.

### 2.2. Characterization of ND/MoS_2_ Nanocomposites

The ultraviolet (UV) absorption of ND-COCl, MoS_2_ nanosheet, and ND/MoS_2_ nanocomposite was measured with a V-670 UV-Vis/NIR spectrophotometer (JASCO Corp., Tokyo, Japan). These measurements were performed under the following conditions: scanning speed: 200 nm/min, data interval: 1 nm, UV-Vis bandwidth: 1.0 nm, near-infrared (NIR) bandwidth: 2.0 nm, and wavelength: 350–900 nm [29]. The dispersions (0.2 mg/2 mL) of the nanosheets and the nanocomposite in deionized (DI) water were placed, respectively, in a synthetic quartz cuvette (light path: 1 cm, Hellma Analytics, Müllheim, Germany). Measurement of Fourier transform infrared spectroscopy (FT-IR; Nicolet 6700, Thermo Scientific., Waltham, MA, USA) ranged from 4000–500 cm^−1^ of wavelength and was performed subsequently on the chemical reactant, ND, and MoS_2_. IR samples were prepared using the standard method of KBr pellet (7 mm in diameter). The materials of 0.5–1 mg were added with a portion of KBr (7 mg). 

X-ray photoeletron spectroscopy (XPS; ThermoFisher Scientific Co., Waltham, MA, USA) measurements were also conducted with an Al Kα energy source. The spectra were analyzed using Avantage software (version 1.6, Thermo Fisher Scientific, Waltham, MA, USA). Transmission electron microscopy (TEM) images were obtained with a JEM-2100F electron microscope (JEOL, Tokyo, Japan). For sample preparation, 10 μL each of ND, MoS_2_, and ND/MoS_2_ nanocomposite was dropped on Formvar/Carbon on a 200 mesh grid (TED PELLA Co., Redding, CA, USA.) and dried for 10 min at 60 °C in an oven. 

### 2.3. Cyclic Voltammetry (CV) Measurements

A dispersion of ND/MoS_2_ nanocomposite (1 mg/mL) was prepared in Nafion solution (0.5% in DI water) for electrode fabrication. Droplets of the dispersion (10 μL) were placed on the glassy carbon working electrode (diameter 3 mm) and dried at 80 °C for 1 h. Cyclic voltammetry (CV) measurements (potential: −0.8 to 0.2 V, scan rate: −0.2 to 1.0 V) were performed using a three-electrode system (reference electrode: Ag/AgCl, counter electrode: platinum wire (57 mm in length, OD 0.5 mm), working electrode: ND/MoS_2_ or MoS_2_ deposited glassy carbon electrode). Two sets of measurements using a potensiostat (DY2322, Digi-Ivy, Austin, TX, USA) were also performed at potentials ranging from 0.05 to 0.5 V·s^−1^ and scan rates of 0.05, 0.1, and 0.5 V/s. The corresponding CV plots were recorded in a 0.1 M KOH solution (15 ml). Five cycles in 2 sets were performed to obtain a voltamogram. After stabilization from repeated cycles, an oxidation and reduction curve in each second set of a working electrode was shown for data presentation. 

## 3. Results and Discussion

UV–vis absorption spectra were obtained for the MoS_2_ nanosheets, ND-COCl, and ND/MoS_2_ nanocomposite with ratios of 1:1, 1:2, 1:4, and 1:8 (Figure 1). Typical MoS_2_ excitation absorption peaks, which occurred at 630 and 690 nm, were attributed to the direct gap transitions at the K point [30,31,32]_._ These indicate the lowest optical band gap of the MoS_2_ nanosheets (i.e., ~1.8 eV) that were changed, owing to the quantum confinement in the sheets. Changes in the UV absorbance were evaluated for the MoS_2_/ND nanocomposite with ratios of 1:1 to 1:8. The optical absorbance peaks of MoS_2_/ND nanocomposite appeared in the same region as those of the MoS_2_ nanosheets. It was indicated that chemically conjugated MoS_2_/ND nanocomposite maintained the optical characteristics of the nanosheets. The intensity of the bands in the spectra were augmented with the increasing ratio of the nanocomposite, as shown in Figure 1 [33]. 

Chemical conjugation of the nanocomposite and functional groups of intermediate products was characterized via FT-IR spectroscopy to identify sequential chemical modification of the acyl chloride comprising ND (ND–COCl), MoS_2_-NH_2_, and MoS_2_/ND nanocomposites (Figure 2). The oxidative treatment formed a carbonyl group (C=O) and a hydroxyl group (-OH), occurring as absorption peaks at 1718 and 3400 cm^−1^ on the surface of the carboxylated ND, respectively. The graphitic shell around the crystalline ND was attributed to the absorption bands associated with C=C bond bending at 1625 cm^−1^. After acyl chlorination, ND with an acyl chloride bond (C-Cl) was formed, corresponding to the stretching peak at 593 cm^−1^, whereas no peak for ND-COOH was observed [34]. The results indicated that the carboxylated ND surface was activated by the acyl chloride group for an amine reactive reaction. The functionalization of MoS_2_ with cysteamine hydrochloride was characterized with comparison of the cysteamine hydrochloride. For the amine-functionalized MoS_2_, an N−H deformation vibration peak and an NH_2_ stretching vibration peak, derived from the chemical conjugation with cysteamine hydrochloride, occurred at 1604 and 3390 cm^−1^, respectively [35]. This indicated the successful modification of amine-functionalized MoS_2_, where amine groups were positioned on the surface of MoS_2_ nanosheets. The formation of an amide bond (NHCO) between the acyl chloride of ND and the functionalized MoS_2_ nanosheets was confirmed via FT-IR spectroscopy of the ND/MoS_2_ nanocomposites. The spectra obtained for the nanocomposites exhibited characteristics of both ND-COCl and the functionalized nanosheets. The absorption bands associated with the ND/MoS_2_ peak were attributed to hydroxyl bond stretching and an amide bond (NHCO) at 3400 cm^−1^ and 1632 cm^−1^, respectively. Furthermore, the relatively strong absorbance band region resulted probably from the overlapping of bands associated with C=C vibration and C=O stretching [36,37]. 

The chemical structure of the ND/MoS_2_ nanocomposite, compared with that of ND-COCl, was determined from the measured XPS spectra, as shown in Figure 3a–f. The C1s peaks of ND-COCl and ND/MoS_2_ were decon-voluted into four component peaks using Gaussian fitting at 284.8 (C-C), 286.0 (C-O), 287.2 (C=O), and 289 (-COOH) eV, respectively [38,39]. The C=O peak at 287.2 eV for ND-COCl was generated from both acyl chloride and some portion of the carboxyl group. The C=O peak at 287.2 eV arose from the amide group after chemical conjugation of ND/ND/MoS_2_ nanocomposites. After chemical conjugation between ND-COCl and amine-functionalized MoS_2_, the intensity of the peak at 286.0 V (C-O) increased significantly relative to that of the peak at 287.2 V (C=O)_._ The results indicate that unreacted acyl chloride would decompose into carboxyl groups during the washing process and from contact with moisture. The O1 peaks of ND-COCl and ND/MoS_2_ at 531.8 and 533.1 eV were attributed to C=O and COC/COH, respectively. The intensity of the ND/MoS_2_ peak increased relative to that of the ND-COCl peak, indicating that the unreacted acyl chloride group was converted to the carboxyl group after chemical conjugation with the MoS_2_ nanosheets (Figure 3b,e). The Cl 2p peaks of the ND-COCl were deconvoluted into two conventional binding energies of 200.3 eV (Cl 2p _3/2_) and 201.9 eV (Cl 2p _1/2_). This indicated that ND-COOH was functionalized by an acyl chloride group on the surface [36]. Deconvolution of the Cl 2p peaks, corresponding to ND/MoS_2_ generated peaks at 200.3 eV (Cl 2p _3/2_) and 201.9 eV (Cl 2p _1/2_), have resulted from the physical adsorption of remnant chloride ions onto the ND surface and nanocage [34,36].

Figure 4 shows the XPS spectra of the MoS_2_ and ND/MoS_2_ nanocomposite. Two strong peaks for the Mo 3d peak of MoS_2_ are observed at 229.85 for doublet Mo 3d_5/2_ and 232.98 eV for and Mo 3d_3/2_ (Figure 4a). The peaks, corresponding to the S 2p_1/2_ and S 2p_3/2_ orbital of divalent sulfide ions (S_2_−) occur at binding energies of 163.85 and 162.65 eV, respectively, as shown in Figure 4b. The results are consistent with the values reported for a MoS_2_ crystal [35,40,41,42]. The XPS spectra of ND/MoS_2_ also revealed typical MoS_2_ crystalline characteristics with chemically induced shifting. Furthermore, the peak position moved from 229.85 to 229.53 eV for Mo 3d_5/2_ and from 232.98 to 232.64 eV for Mo 3d_3/2_, respectively. Chemically induced shifts were also observed for ND/MoS_2_ S 2p, with the peaks shifting from 162.65 to 162.44 eV for S 2p_3/2_ and 163.85 to 163.71 eV for S 2p_1/2_ peaks [43]. This indicated that the chemical shift of the ND/MoS_2_ nanocomposite to lower binding energy than that of MoS_2_ resulted from chemical conjugation with ND. 

The morphologies of MoS_2_, ND, and ND/MoS_2_ nanocomposites (Figure 5) were evaluated with TEM. The MoS_2_ nanosheets was featured as average size of 300–400 nm along the long axis (Figure 5a,b). The MoS_2_ nanosheets had a crystalline structure, and several molecular layers were stacked or folded in a planar form [44]. Furthermore, the NDs were well-dispersed with small agglutinins ranging from 80 to 200 nm (Figure 5c,d). The ND/MoS_2_ nanocomposite with chemical conjugation showed morphological features that several molecular layers of thin MoS_2_ nanosheet enveloped ND agglutinins (Figure 5e–h). Amine-functionalized MoS_2_ nanosheets and NDCOCls were chemically reacted through surface contact, suggesting that the ND/MoS_2_ nanocomposite was successfully synthesized (Figure 5e–h).

Cyclic voltammograms of the MoS_2_ nanosheet and ND/MoS_2_ are shown in Figure 6a–f. For both electrodes, the rectangularity of the CV plots decreased slightly, corresponding to the reversible reactions of Mo^2+^/Mo^3+^ associated with OH^-^ anions [45]. The possible redox reaction is given as follows [46]:
ND/MoS_2_ + OH^- ↔^ ND/MoSOH + e^-^
ND/MoSOH + OH^- ↔^ ND/MoSO + H_2_O + e^-^

On the same material set, scan rates of 0.05, 0.1, and 0.5 V/s, respectively, and range of potential voltage was differently applied to −0.8 to −0.2 V and −0.2 to 1.0 V. The shape and magnitude of the voltamogram was transitioned from a peak-like shape (Figure 6a–c) to quasi-rectangular shape (Figure 6d–f) depending on the scan rate and potential voltage. The quasi-rectangular shape typically indicates constant and time dependent concentration gradient of an electroactive surface where the electrode radius was typically smaller than diffusion layer [47]. Each CV graph is characterized by a quasi-rectangular shape, consistent with dual behavior, such as the electrical double layer capacitance [48]. MoS_2_ nanosheets constitute the minimum area of the CV plot, whereas 1:8 and 1:6 ND/MoS_2_ constitutes the maximum area, which corresponds to the enhanced capacitance. The CV curve of the ND/MoS_2_ nanocomposites reveals the higher current response and large working area of these composites, compared with those of the MoS_2_ nanosheet only (Figure 6). The results suggest that the addition of nanodiamonds enhances the electrochemical activity and increases the specific capacitance of MoS_2_ electrode alone [29,35]. 

The superior electrical performance of the ND/MoS_2_ nanocomposite electrode, compared with that of the MoS_2_ electrode alone, resulted from the unique nanostructure of the composite electrode. Moreover, the large surface area and the nanosized MoS_2_ phase of the ND/MoS_2_ composites may have resulted in significant reduction of the diffusion length associated with ion and electron transfer during the oxidation/reduction process. The electrode nanoscale phase makes these composites promising for various applications. Specifically, the NDs acted as nanoscale supports to functionalize synergistically the MoS_2_ sheets, which served as a three-dimensional highly conductive current collector. The featured architecture of the ND/MoS_2_ nanocomposites possessing a large specific surface of the electrode enables rapid and simultaneous electron and ion transport, thereby leading to excellent electrochemical capacitive performance [42]. 

## 4. Conclusions

A chemically conjugated ND/MoS_2_ nanocomposite was synthesized with amine-functionalized MoS_2_ and acyl chloride-coordinated NDs. The chemical structure and morphology of the nanocomposite were characterized, and the results revealed that the MoS_2_ nanosheets were well-distributed on the ND platform, thereby forming a nanophase. Nanophase distribution of MoS_2_ on ND with a graphitic shell may provide a large surface area and reduce the diffusion distance of ions and electrons. Therefore, the augmented electrochemical capacitance of the nanophase electrode was induced, compared to that of the MoS_2_ electrode alone.

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
