# Peer review of "Nanocomposite Synthesis of Nanodiamond and Molybdenum Disulfide"

_nanomaterials, 2019, doi:10.3390/nano9070927_

Round 1
Reviewer 1 Report
This manuscript proposes a nanocomposite of nanodiamond and molybdenum disulfide for efficient electrocapacity. The topic is interesting, and certainly consistent with the contents to be proposed to the readers of “Nanomaterials”. Moreover, the manuscript is well written and can be read with pleasure: this represents an important aspect in the current scenario of publications in international journals. Overall, I think that this manuscript has to be accepted, but the Authors should take into account the following minor revisions (in terms of bibliographic updates, grammar corrections and content deepening):
- Detailed revisions: I spent several hours reading this manuscript, and Authors are asked to follow carefully the attached PDF file where I highlighted some points to be addressed. The attached file also contains language mistakes and typos; some questions related to manuscript contents could also be present and Authors must consider them properly before submitting the revised manuscript. A point-by-point reply is required when the revised files are submitted.
- The Introduction should give a wider overview on the present scenario related to nanomaterials based on carbon and transition metal oxides, both in terms of recently published reviews and research articles. In particular, applications in solar cells, batteries and supercapacitors are missing and a paragraph on this topic is highly suggested to be added in the Introduction. Authors are invited to go through the literature published in the last six months on these issues, and also on concepts developed some years ago in this field. Some of them are also mentioned in the above mentioned PDF file.
- Authors should provide a clear explanation on the experimental error of the proposed research work. In particular, reproducibility of the phenomena described in the manuscript should be clearly stated in the “Results and Discussion” section; besides, some notes in the “Materials and Methods” section should be added highlighting which kind of experimental approach has been followed to check the reproducibility of the proposed system, the latter being of noteworthy importance in the present research field.

Author Response
Please see the attachement.

Reviewer 2 Report
With the rapid development in nanoscience and nanotechnology, a large variety of nanomaterials with diverse sizes, compositions and structure characteristics need to be synthesized and applied for potential applications. Molybdenum disulfide (MoS2) is generally considered as an ideal substrate for hybridizing with functional groups, to form MoS2-based nanocomposites. Integrating such nanocomposites with nanodiamonds possess excellent properties due to synergistic effects, therefore greatly broadening the applications of MoS2.
Authors have synthesized ND/MoS2 (pretty much like MoS2 decorated with ND nanocomposites).
Several methods have been reported using noble metal nanostructures in MoS2, however, the ND is not widely reported. The reported electrodeposition is another facile method to synthesize MoS2/ND nanocomposites due to the fact that it is simple, fast, and environmentally friendly. The manuscript needs to be improved before rendering a final decision.
Importantly, authors need to quantify the capacitance value, more elaborate reaction mechanism drawn from CV curves, also providing charge-discharge curves (if available). The electrochemical part is quite weak, which is the selling point of this work. Provide the performance characteristics of the material (like capacitance and the equation used to calculate this).
My other specific comments are below.
· Title: What does the term “electrocapacity” refer? Does the author mean electrochemical performance? Consider revising it.
· Introduction: In the last paragraph, add one of the objectives of this study is to examine the nanocomposites material for electrochemical capacitors.
· Experimental: Please provide the dimension of the Pt wire used in CV.
· Experimental: Please provide the sample preparation for IR analysis.
· The XPS spectra obtained for the Mo 3d peak doublet and its corresponding binding energies need to be referred back to the literature (Dalton Trans. 46 (2017) 3588; ChemPlusChem 81 (2016) 964).
· How does the obtained morphology “nanosheet” compare to that of Mo structure reported in the literature for nanoplate (Nanoscale 10 (2018) 13277)). Cite and discuss the differences in morphology and its synthesis conditions.
· Page 8; lines 205 – 208 – these lines have already said in the experimental section, so better to remove.
· Lines 209 -210: why MnO2 (ref. 38; published in a journal “Carbon” is related to manganese dioxide material) has been shown as a reference? Maybe out of context. Please refer to Mo related material.
· Lines 211- 212: “Each CV graph is characterized by a quasi-rectangular shape, consistent with dual behavior, such as the electrical double layer capacitance” this line need to be referred back to the literature (New J Chem 40 (2016) 7456).
· Line 214: enhanced capacitance – please quantify the value.
· Lines 221 – 222: “charge/discharge” The curves shown in Fig. 6 is a CV study so it should read as “oxidation/reduction process”.
· Fig. 6d, left-hand side, increase the potential window only then the initial CV curve could be viewed. Currently, it cannot be viewed
· Fig. 6 a, b, c, exhibit a peak-like shape appeared at -0.4 V while Fig. 6 d, e, f quasi-rectangular shape; please explain the difference and nature of the material/scan rate.
· In conclusion, please quantify the performance characteristics of this material. Currently, it is missing.
Author Response
Please see the attachement.

Round 2
Reviewer 2 Report
Authors have addressed my queries satisfactorily.